# Endotoxin-Induced Sepsis on Ceftriaxone-Treated Rats’ Ventilatory Mechanics and Pharmacokinetics

**DOI:** 10.3390/antibiotics13010083

**Published:** 2024-01-15

**Authors:** Juliana Savioli Simões, Rafaela Figueiredo Rodrigues, Bruno Zavan, Ricardo Murilo Pereira Emídio, Roseli Soncini, Vanessa Bergamin Boralli

**Affiliations:** 1Faculdade de Ciências Farmacêuticas, Universidade Federal de Alfenas (UNIFAL-MG), Alfenas 371300-001, Brazil; julianasavioli@hotmail.com (J.S.S.); rafaelafigueiredor@gmail.com (R.F.R.); 2Insituto de Ciências da Natureza, Universidade Federal de Alfenas (UNIFAL-MG), Alfenas 371300-001, Brazil; bruno_zavan@yahoo.com.br (B.Z.); ricardo.emidio@sou.unifal-mg.edu.br (R.M.P.E.); roseli.soncini@unifal-mg.edu.br (R.S.)

**Keywords:** sepsis, ARDS, ceftriaxone

## Abstract

Sepsis can trigger acute respiratory distress syndrome (ARDS), which can lead to a series of physiological changes, modifying the effectiveness of therapy and culminating in death. For all experiments, male Wistar rats (200–250 g) were split into the following groups: control and sepsis-induced by endotoxin lipopolysaccharide (LPS); the control group received only intraperitoneal saline or saline + CEF while the treated groups received ceftriaxone (CEF) (100 mg/kg) IP; previously or not with sepsis induction by LPS (1 mg/kg) IP. We evaluated respiratory mechanics, and alveolar bronchial lavage was collected for nitrite and vascular endothelial growth factor (VEGF) quantification and cell evaluation. For pharmacokinetic evaluation, two groups received ceftriaxone, one already exposed to LPS. Respiratory mechanics shows a decrease in total airway resistance, dissipation of viscous energy, and elastance of lung tissues in all sepsis-induced groups compared to the control group. VEGF and NOx values were higher in sepsis animals compared to the control group, and ceftriaxone was able to reduce both parameters. The pharmacokinetic parameters for ceftriaxone, such as bioavailability, absorption, and terminal half-life, were smaller in the sepsis-induced group than in the control group since clearance was higher in septic animals. Despite the pharmacokinetic changes, ceftriaxone showed a reduction in resistance in the airways. In addition, CEF lowers nitrite levels in the lungs and acts on their adverse effects, reflecting pharmacological therapy of the disease.

## 1. Introduction

Sepsis is associated with the progressive deterioration of acute lung injury (ALI) and also acute respiratory distress syndrome (ARDS). The process is initiated by the destructive effects of neutrophils and platelets on the endothelial and epithelial tissues of the lungs, leading to the infiltration of a fluid containing fibrin, red cells, and neutrophils into the alveoli, resulting in edema. Even a tiny amount of substance has a detrimental effect on the surfactant, rendering it inactive. As a result, the collapse of the alveoli occurs, leading to a decline in lung function. The histomorphological abnormalities detected in the organ include bilateral opacity, heightened capillary permeability, elevated pulmonary pressure, weight, pleural petechiae, and hemorrhage [1].

ARDS is reported as a significant clinical problem and the final pathway of lung injury from multiple etiologies [2]. ARDS is an inflammatory state characterized by infiltration of mixed inflammatory cells, diffuse destruction of the alveolar–capillary barrier, and severe edema, consequently causing hypoxemia and increased lung density [3]. In addition, it can be considered a consequence of other critical illnesses such as sepsis or pneumonia [2]. Sepsis syndrome and ARDS are closely related. The principal link between them is the sepsis syndrome, the commonest precipitating cause of ARDS; septic ARDS has a higher mortality than the ARDS of other aetiologies [4]. Clinical trials have yet to find a therapy to decrease morbidity and mortality.

Sepsis and ARDS together can cause endothelial damage in veins, microcirculation, and capillary extravasation due to the induction of the vascular endothelial growth factor (VEGF) and nitric oxide (NO), modifying the volume of distribution and, consequently, the pharmacokinetics of antibiotics such as ceftriaxone [5].

The recovery of patients is influenced by several factors, with the selection of antimicrobial therapy and the correct dose being key determinants [6,7]. The dosage given to the patient is determined by pharmacokinetic tests conducted on healthy and non-obese persons. Nevertheless, the clearance of antibiotics can be affected by changes in glomerular filtration, which can be attributed to factors such as multiple organ failure and other variables that impact the pharmacokinetics of these drugs. β-lactam antibiotics, including cefepime and ceftriaxone, can potentially experience changes in their pharmacokinetics [8].

Ceftriaxone (CEF) is a cephalosporin antibiotic with potent antimicrobial activity against the common Gram-negative and Gram-positive bacteria [9]. The stability and relatively long elimination half-life of ceftriaxone (~7 h) make this cephalosporin a good choice for the treatment of patients with several infections, including intra-abdominal infections in humans [10,11] and animals [12]. This antibiotic, a third-generation cephalosporin, has been approved by the World Health Organization (WHO) as the primary antibiotic of choice for managing patients admitted to intensive care units [13]. The utilization of this substance in the treatment of systemic infections is attributed to its prolonged elimination half-life and efficient penetration into multiple body fluids [14]. The pharmacokinetics of the cephalosporin are influenced by the pathophysiological changes associated with sepsis, leading to alterations in both the volume of distribution and clearance. These changes could impact the concentration of the free fraction and thus compromise the therapeutic efficacy of the drug [15,16].

The endotoxin lipopolysaccharide (LPS) is a well-recognized inducer of ALI and ARDS [17]. This same author reported that endotoxin promotes evident and subtle effects on airways and pulmonary circulation functions. The effect of LPS on the pulmonary alveoli is complex. First, endotoxin causes lung damage by interacting with individual types of lung cells and initiating the secretion of various inflammatory mediators. Subsequently, this process results in the disruption of alveolar epithelial and endothelial barriers. Second, LPS also interferes with the ability of alveolar type II cells to produce pulmonary surfactant, and LPS itself binds to surfactant proteins and phospholipids, leading to surfactant inactivation [18].

Experimental animal models facilitate the comprehension of pathophysiological alterations by providing the ability to manipulate independent factors. Hence, it is possible to quantify and comprehend variables that are influenced by others, and by establishing a cause-and-effect connection, it becomes viable to assess the hypotheses put forth [19]. ALI/ARDS can be experimentally generated in animals through many methods, such as sepsis, transfusion, multiple trauma, aspiration of gastric contents, and tissue reperfusion ischemia [20,21]. ARDS due to sepsis is initiated by alterations in surfactant levels, the activation of the innate immune system response, changes in coagulation, the presence of cells within the alveoli, and damage to the endothelium glycocalyx [22].

However, the alterations in the parameters of respiratory mechanics and the precise mechanisms behind these changes still need to be better comprehended. Even though beta-lactams are the recommended treatment, there needs to be more understanding regarding their impact on breathing mechanics.

Once infections are the most common etiology of ARDS and cephalosporin was used to improve the disease outcome [23], we hypothesized that ARDS could cause physiological changes that can alter treatment efficacy. Therefore, in this study, we integrated an investigation of the pharmacokinetics of ceftriaxone and its impact on the pulmonary function, aiming to enhance comprehension of the treatment effectiveness in ARDS.

## 2. Results

The experimental design was defined to study the CEF consequences on the lungs after one dose administration and one hour of acute activity (close to the maximum plasma concentration (Cmax) of CEF in rats). All groups analyzed exhibited similar values on the respiratory mechanic’s parameters after aerosolized saline (baseline). Forward was aerosolized methacholine (MCh) at 50 mg/mL to assess bronchoconstriction and evaluate the ability of the response of the respiratory systems to a challenge. The values registered to the MCh 50 mg/mL were not different in all the studied groups (Figure 1).

However, after increasing the MCh dose (aerosolized MCh 100 mg/mL), the response was explicitly reduced on Raw in the control + CEF and LPS + CEF groups (0.443 ± 0.085, *p* < 0.01 and 0.375 ± 0.039, *p* < 0.001; respectively) in comparison to the control group (0.680 ± 0.081 cm H_2_O·s·mL^−1^). This result indicates that, even in just one hour, the CEF improves the pulmonary functions shown by Raw.

The BALF inflammatory cells and NOx values are shown in Table 1. Analysis of bronchial lavage assessed the amount of inflammatory cells present in the bronchoalveolar environment. The data obtained indicate that the administration of one dose of ceftriaxone via the intraperitoneal route results in a significantly higher number of inflammatory cells in the alveolar space after one hour compared to the control group (842.81 vs. 456.56; *p*: 0.0064). Similar results are seen following the introduction of endotoxin (1135.00 vs. 456.56; *p*: 0.0006) and its combination with the antibiotic (LPS + CEF group) (1194.68 vs. 456.56; *p*: 0.0017). The data illustrate the antibiotic’s capacity to recruit inflammatory cells to the bronchoalveolar environment.

A substantial increase in macrophages was observed in the LPS group compared to the control group (61.52 vs. 48.38; *p*: 0.0022). The rats in the LPS + CEF group exhibited a decrease in the values of these cells compared to the LPS group (53.96 vs. 61.52; *p*: 0.0022). The CEF group has lower macrophage levels than the LPS group (43.84 vs. 61.52; *p*: 0.0003).

Upon evaluating the lymphocyte values, it is evident that there is an increase in levels in the LPS groups (44.46 vs. 35.34; *p*: 0.0281) and LPS + CEF (43.89 vs. 35.34; *p*: 0.0070) compared to the control group. The CEF group showed a significant increase compared to the control group (52.83 vs. 35.34; *p*: 0.0030).

A decrease in values of foam cells was observed in animals belonging to the CEF group compared to the other groups. However, a statistically significant difference was only found when comparing the CEF group to animals from the LPS group (2.03 vs. 3.86; *p*: 0.0281) and the control group (2.03 vs. 2.86; *p*: 0.0281).

Neutrophil levels were higher in the groups treated with endotoxin. However, the groups who received the antibiotic together with the endotoxin (LPS + CEF group) showed statistically significant differences compared to the animals in the control group (2.25 vs. 0.44; *p*: 0.0353) and the CEF group (2.25 vs. 0.38; *p*: 0.0174).

In the groups treated with endotoxin, there was a notable rise in the overall levels of inflammatory cells. However, the antibiotic could not wholly reduce the quantities of inflammatory cells in the BAL.

The levels of the inflammatory marker NOx were significantly higher in all groups compared to the control group 24 h after the condition induction by endotoxin. Additionally, the impact of CEF on sick animals is clear, as the nitrite levels in the LPS + CEF group were significantly lower than in the LPS group (6.82 vs. 15.75 μM, *p*: 0.0002). Nevertheless, ceftriaxone successfully stimulated the production of NOx, as evidenced by the higher levels found in the CEF group compared to the control group (3.59 vs. 2.31 μM; *p*: 0.0030). The LPS group presented elevated values, which were subsequently decreased in the LPS + CEF group. However, these values remained greater than those observed in the control and control + CEF groups (*p* < 0.01).

The expression of VEGF protein in this method was measured by quantifying the number of cells compared to the overall area (cell number/area) and the airways (pixel/airway area deducted from the lumen), which were stained positively by immunohistochemistry (areas showing a brown color). In the histological sections of the airways, the expression of VEGF is higher in animals in the LPS group compared to the other groups. The values for VEGF expression in pixels per luminal were 606.3 for the control group, 386.1 for control + CEF, and 382.5 for LPS + CEF. However, the administration of CEF in the LPS + CEF group reduced this increase in VEGF expression. The difference in VEGF expression between the LPS group and the other groups is statistically significant (*p* ≤ 0.01). The results for the airways can be shown in graph form and photomicrography in Figure 2.

Furthermore, the LPS group exhibited a decrease in the airway luminal area, compared to the control group, which was subsequently repaired by ceftriaxone treatment, as shown in the LPS + CEF group.

The pharmacokinetic parameters are shown in Table 2. The LPS + CEF group exhibited a statistically significant reduction in the area under the curve parameter compared to the treated animals (control + CEF) (514.01 vs. 672.47, *p:* 0.0002). Furthermore, differences in clearance were observed, displaying a divergent pattern compared to the results obtained from the analysis of the area under the curve. Animals with sepsis had higher values than the healthy animals (195.30 vs. 149.31 mL/h/kg), which was statistically significant with a *p*-value of 0.0002. The LPS + CEF group substantially increased the elimination constant parameter (103.33%; *p*: 0.0401).

Nevertheless, the elimination half-life noticed a significant decrease of 67.86% (*p:* 0.0002). Similarly, the time required to reach the maximum plasma concentration (Tmax) and the maximum plasma concentration (Cmax) also showed a notable reduction of 32.07% (*p*: 0.0002) and 16.04% (*p*: 0.0002), respectively.

In Figure 3, the effects of sepsis on the analyzed pharmacokinetic parameters can be observed by superimposing the mean plasma concentration curves as a function of time.

A decrease in bioavailability (AUC^0–∞^) was observed in rats receiving endotoxin 24 h before antibiotic administration (514.01 vs. 672.47 µg×h/mL) with a *p* value of 0.0002. The LPS + CEF group had higher clearance values than the CEF group (195.30 vs. 149.31 mL/h/kg) with a *p*-value of 0.0002. Increased clearance in the LPS + CEF group resulted in a vital process speed reduction, with the elimination rate (t_1/2_) decreasing by 67.86% (*p*: 0.0002). A similar effect was seen for Tmax and Cmax, which dropped 32.07% (*p*: 0.0002) and 16.04% (*p*: 0.0002).

## 3. Discussion

Gram-negative bacterial infections induce endotoxemia, which is excess LPS in the blood; infection causes widespread inflammation. The present study observed sepsis after 24 h of LPS (IP) administration. It is based on previous studies [24] and agrees with our expertise in other protocols for ARDS-induced [25]. To determine the effects of CEF administration on pulmonary functions following bronchoconstriction challenge and BALF inflammatory markers, animals treated with LPS and controls were investigated.

A common approach to assessing respiratory mechanics and measuring airway responsiveness is measuring the parameters. This refers to the ability of the routes to be reduced considerably after administering a constriction agent, such as methacholine. The bronchoconstrictor provocation test is employed in airway responsiveness assessments to induce an enhanced response to the medication, resulting in increased reactivity within a short time [26].

Gram-negative bacteria-derived LPS targets the lung. Overactivation of the immune system by LPS or other inflammatory stimuli can modify physiological characteristics such as pulmonary vascular resistance, bronchoconstriction, and airway hyperreactivity, causing ARDS [27]. Raw fluctuates by lung capacity, disease, and inflammation [28].

Raw is verified by calculating the ratio of the airway pressure differential to the airflow rate. The measure can fluctuate based on lung capacity, existing disease, and inflammatory response [28]. The observation supports the study’s results, demonstrating a decrease in the Raw parameter in both the LPS and LPS + CEF groups. This anticipated outcome suggests a complicated relationship between muscle control activity and acute neutrophilic inflammation [29]. Nevertheless, the control + CEF group decreased the airway resistance, comparable to the LPS-treated group, indicating that the antibiotic enhanced airflow by reducing resistance after one hour of dosing.

The immune response in sepsis depends on the progress of the pathology. Initially, there is a visible increase in the number of inflammatory cells caused by the activity of monocytes that stimulate the formation of these cells. A decrease in the number of inflammatory cells and a reduction in monocyte activity are found as the illness advances [30]. At first, there is an initial systemic inflammatory reaction, followed by compensating negative feedback [31].

Total inflammatory cells increased significantly in LPS-treated groups. Previous studies have also found increased inflammatory markers [32]. This may be caused by the LPS groups’ increased NO and VEGF production, which changes endothelial permeability. As a result of being administered in a single dosage and just an hour after administration, the antibiotic could not reduce BAL inflammatory cell numbers. Current studies established a clear relation between LPS administration and VEGF expression in the lung [33,34] in the long term. These articles indicate that the VEGF impairments could be a possible target to damage lungs induced by LPS instillation in mice. Our study shows an increase in VEGF expression in the airway; however, we also demonstrated a reduced luminal area in a short time.

These findings are consistent with other previously published work [35], which discovered elevated inflammatory activity in BAL from septic rats and increased neutrophil and macrophage recruitment rates. The decrease in neutrophils seen in the CEF group compared to the control group has already been published in the literature with statistical significance in patients with Lyme disease treated with ceftriaxone for 12 days [36].

The VEGF affects the properties of endothelial cells, such as NO and prostacyclin production. The NO is critical to the pulmonary circulation and mediates the permeability effects of the VEGF; this way, the concentrations observed for NO may be temporally linked to the progression of septic shock [37]. In the lung vessel, the VEGF, prostacyclin, and NO are always intricately linked and form the pillar of their biology [38]. In asthma, the VEGF potentiates inflammation, airway remodeling, and physiologic dysregulation [39].

Our investigation found that animals treated with endotoxin had higher NOx levels than other groups; the LPS group exhibited higher values. Although antibiotic treatment lowered these values for the LPS + CEF group, they remained higher than the control and CEF groups. The elevated concentrations of NOx observed in our study in animals with sepsis indicate changes in the permeability of blood vessels in the lungs. These changes could be associated with an increase in the total number of leukocytes in BAL samples.

LPS is a major component of the outer membrane of Gram-negative bacteria and is known to be a key pathogenic stimulator for multiple organ dysfunction. In sepsis, circulating LPS as a pathogen-associated molecular pattern (PAMP) stimulates the innate immune system and mediates local or systemic inflammation. This way, we believed the LPS-induced ARDS and NOx in the BALF are increased, suggesting that they were triggered by the inducible NO synthase (iNOS), as previously described in the literature [40].

Sepsis patients may have capillary permeability and endothelial lesions in veins and microcirculation due to VEGF production. Prolonged mechanical respiration increases oxygen levels and reduces the VEGF in glandular cells and alveolar epithelial tissue, decreasing pathology-related VEGF production [1]. Septic patients had higher airway VEGF expression. In our study, CEF decreased the VEGF in the LPS + CEF and control groups.

Ceftriaxone is very stable against most beta-lactamases, including cephalosporinases and penicillinases, and against many Gram-positive and Gram-negative pathogens. Unlike other cephalosporins, this treats severe infections caused by multidrug-resistant Gram-negative bacteria and bacterial meningitis. Drug monitoring is recommended due to the significant variation in response and severity of the circumstances it can treat [41]. The patient’s clinical status can greatly influence pharmacokinetics. A literature review [42] found that the beta-lactam volume of distribution and clearance are often changed in critically ill patients, with higher variability than in healthy people.

Compared to the control group, animals treated with LPS had higher clearance values and lower bioavailability. Our study’s VEGF and NO findings may explain this result. In septic male rats treated with ceftriaxone, the authors observed that the glomerular filtration barrier lost permeability and selectivity, causing albuminuria. Sepsis eliminates ceftriaxone, which is highly bound to plasma proteins [43].

In our investigation, animals developed sepsis after 24 h. Ceftriaxone reduced the resistance, improving the airflow. Reducing the lung VEGF expression and nitrite levels reduces infection and suggests pharmacological action of the CEF. Our data indicated that the indicators improved despite decreased CEF bioavailability for septic animals.

## 4. Materials and Methods

### 4.1. Animals

All procedures performed in the present study comply with the defined guidelines of the institutional animal care committee (ethical approval protocol by number 05/2016). The adult male Wistar rats (250 ± 50 g) were housed at 22 ± 2 °C and maintained in a 12:12 h light-dark cycle. The animals were housed in polypropylene cages with woodchip bedding and were fed with standard chow and water ad libitum.

### 4.2. Experimental Outline

For the study, the animals were randomized into four main groups: Control, control + CEF (animals treated with ceftriaxone), LPS (animals treated with lipopolysaccharide), LPS + CEF (animals treated with lipopolysaccharide and ceftriaxone), and underwent respiratory mechanics tests, BAL analysis, VEGF and ceftriaxone analysis. The experiments were not the same for all groups, with respiratory mechanics, BAL, and VEGF analysis being the four groups (n = 8). The pharmacokinetics of ceftriaxone were only evaluated with the CEF and LPS+ CEF groups (n = 8). This way, for the control, we have three groups (respiratory mechanics, VEGF, BALF); we have the same three groups for LPS. For control + CEF, we have four groups (respiratory mechanics, VEGF, BALF, and pharmacokinetics), and the same four groups were presented for LPS + CEF. In total, we have 14 different groups of rats.

The LPS used is from Escherichia coli serotype 026:B6, obtained from Sigma Chemical (St. Louis, MO, USA). The dose of LPS used was 1 mg/kg [44]; for CEF, the dose was 100 mg/kg [45], and both substances were intraperitoneally administered, dissolved in sterile saline (NaCl 0.9%), in a final volume of 0.5 mL. The design was: (1) in the control group, the rats received saline. (2) In the control + CEF group, the rats received saline and after 24 h were injected with CEF. (3) In the LPS group, the animals received an LPS injection, 1 mg/kg, and saline IP. (4) In the LPS + CEF group, the rats were injected with LPS and, after 24 h, received CEF.

The beginning of the experimental protocols, time zero, was defined as the point at which the rats received saline or the LPS injection. Later (after 24 h), the animals received a CEF or saline injection, so in the next hour (acute response), the respiratory mechanics and tissue extraction measurements were performed. The pharmacokinetics of ceftriaxone were only evaluated with the control + CEF and LPS + CEF groups (n = eight per group for pharmacokinetics).

The design was: (1) in the control group, the rats received saline (500 µL, IP). (2) In the control + CEF group, the rats received saline (500 µL, IP), and after 24 h, they received CEF. (3) In the LPS group, the rats received LPS, and (4) in the LPS + CEF group, the rats received LPS, and after 24 h, they received CEF.

For physiological alteration studies, the beginning of the experimental protocols, time zero, was defined as the point at which the rats received saline or LPS. Later (24 h after the sepsis-induction), the animals received CEF, and, so, over the next hour (acute response), the measurements of respiratory mechanics, tissue extraction, and alveolar bronchial lavage extraction were performed (n = eight per group).

The pharmacokinetics of ceftriaxone were evaluated only with the control + CEF and LPS+ CEF groups (n = eight per group).

### 4.3. Respiratory Mechanics

The animals initially received anesthesia (pentobarbital sodium, 40 mg/kg, IP, and xylazine, 8 mg/kg, IP). They were then subjected to tracheostomy using an 18-gauge metal IV adaptor and connected to a small animal ventilator (flexiVent, SCIREQ, Montreal, QC, Canada). The ventilation parameters were set to a 6 mL/kg tidal volume, 110 breaths/min, and 3 cm of H_2_O-positive end-expiratory pressure. Before conducting the respiratory mechanics study, the animals were provided with pancuronium bromide (0.5 mL/kg, intraperitoneal—IP) and tramadol (50 mg/kg, intramuscular—IM). Additionally, a heating pad was used to maintain the animals’ temperature. The impedance of the pulmonary system (Zrs) was assessed by introducing a 3-s oscillating volume disturbance to the tracheal cannula, which is attached to the airway opening. The constant phase model [44] was used to fit the acquired data and estimate the mechanical characteristics of the respiratory system, including total resistance (Rtot), compliance (C), elastance (E), airway resistance (Raw), tissue damping (Gtis), and tissue elastance (Htis). This method was specifically developed to quantify the input Zrs in small animals and has previously been comprehensively explained [46,47]. The trials were carried out while maintaining the integrity of the chest wall.

The degree of bronchoconstriction in each rat was evaluated. A solution containing sodium chloride (NaCl) at a concentration of 0.9% and methacholine (MCh) at concentrations of 50 mg/mL and 100 mg/mL, respectively, was transformed into an aerosol form using an ultrasonic device (Aeroneb, Aerogen, Ireland) for 10 s. Respiratory mechanics were evaluated for 90 s, starting 15 s into the assessment. To establish consistent lung volume records, the lungs were inflated once to a pressure of 30 cmH_2_O (a process known as the recruitment maneuver) before the commencement of solution administration. The respiratory mechanics parameters were measured at 30, 60, and 90 s for each rat. Only the data with a coefficient of determination values more than or equal to 0.85 were included. Upon completion of the experiment, the animals were killed while under the effects of anesthetic.

### 4.4. Bronchoalveolar Lavage Fluid (BALF) Cells and Nitrite/Nitrate Concentration (NO_x_)

The BALF was obtained from the same rat as the blood sample, but in separate animals, to evaluate the respiratory mechanics. Following the collection of the blood sample, a tracheostomy was performed on the rat to obtain the lavage fluid. During this technique, a volume of 2.5 mL of phosphate-buffered saline, with a pH of 7.4 (PBS), was introduced into the trachea. The concentration of white blood cells (WBC) in the BALF was determined using a hemocytometer chamber, and their viability was evaluated by eliminating the presence of trypan blue dye. The differentiation of BALF cells was assessed from preparations stained with May–Grünwald/Giemsa. The differential cell count (%) was determined by counting at least 200 cells from each BALF sample.

The nitrite/nitrate content (NO_x_) in the BALF was assessed using the Griess reaction [48] as a means of indirectly quantifying nitric oxide. Overall, a volume of 90 µL was added to a 96-well plate and allowed to incubate at room temperature for 3 h with the addition of 10 μL of nitrate reductase (NADPH Sigma-Aldrich) at a concentration of 0.7 U/mL. Afterward, 50 microliters of a 1% solution of sulfanilamide and 50 microliters of a 0.1% solution of naphthyl ethylene diamine were introduced into each well and allowed to incubate for 5 min at ambient temperature. The microplate reader (Synergy H1, hybrid reader, Biotek, EUA) was used to measure the absorbance of the mixture at 540 nm. The obtained absorbance values were then compared to a standard nitrite curve.

### 4.5. VEGF Immunohistochemistry Analysis

Following the collection of the bronchoalveolar lavage fluid (BALF), the right lung was immersed in a 10% paraformaldehyde solution for fixation and subsequently embedded in paraffin. Seven sections, each with a thickness of one micron, were treated to remove the paraffin and then rinsed with PBS. The sections were treated with 1% hydrogen peroxide (Sigma, St. Louis, MO, USA) and 1% PBS/BSA (Sigma, St. Louis, MO, USA) for 30 min each, followed by overnight incubation at 4 °C with the anti-VEGF-a antibody (Abcan^®^, ab46154) at a dilution of 1:100. The sections had been washed using PBS, then treated with a biotinylated anti-rabbit antibody at a dilution of 1:500 (Sigma, St. Louis, MO, USA). Subsequently, they were exposed to streptavidin-horseradish peroxidase (SA-5704, Vector Laboratories, Burlingame, CA, USA) for 1 h each. Following the PBS wash, the sections were treated with 3.3-diaminobenzidine (Sigma, St. Louis, MO, USA) in Tris-HCl buffered saline pH 7.5 (TBS) 50 mM solution, which also included 0.1% hydrogen peroxide. The sections were stained with Harris’s hematoxylin, mounted with Entellan (Merck, Darmstadt, Germany), and examined using light microscopy (Nikon Eclipse 80i, Tokyo, Japan). A negative control was achieved by excluding the main antibody. A minimal number of five pictures acquired at a magnification of 100× were examined for pixel intensity using the GNU Image Manipulation Program (GIMP 2.8.10 software).

### 4.6. Pharmacokinetic Study

Only two groups were evaluated for pharmacokinetic evaluation: the control + CEF and LPS + CEF (n = eight per group). The animals in the LPS + CEF group were administered lipopolysaccharide (LPS) intraperitoneally (IP) at 1 mg/kg. On the other hand, the control + CEF group received a saline solution intraperitoneally (IP) at a volume of 500 μL. After 24 h, all animals from both groups were subjected to anesthesia using a combination of xylazine (10 mg/kg) and ketamine (75 mg/kg) dissolved in sterile saline and administered intraperitoneally (IP). To introduce the intravenous catheter (24G) into the tail vein of the animal, a Luer-lock connection (BD^®^) was attached to it.

To mitigate the potential impact of catheter insertion on the pharmacokinetics of the antibiotic, a single dose of ceftriaxone (100 mg/kg) dissolved in sterile saline was administered intraperitoneally to the animals 40 min after the cessation of sedation [45].

Blood samples of 500 µL were obtained at specific intervals, at 15, 30, 60, 90, 120, 240, 300, and 360 min [49] from the tail vein in heparinized tubes. Following the collection of blood samples, volume replacement was carried out using sterile saline by intradermal administration. After the last blood collection, the animals were anesthetized and euthanized.

The samples were collected in heparinized tubes and centrifuged (2500× *g* for 10 min). The plasma was separated and stored at −75 °C for the ceftriaxone quantification assay.

Ceftriaxone was measured using high-performance liquid chromatography with a diode array detector (HPLC-DAD). The methodology for the sample preparation followed the previously published protocol [50]. Briefly, 100 µL of a sodium acetate buffer solution 0.7 mM, pH 5.7, was combined with 500 µL of acetonitrile. The resultant solution was subsequently added to 100 µL of plasma. The mixture was agitated using a vortex mixer for 1 min, followed by centrifugation at 2350× *g* for 15 min. Ultimately, a volume of 620 µL of the supernatant was carefully transferred to the designated vial, and a quantity of 20 µL was then injected into the chromatographic system. The separation process employed a C18 column (150 mm × 4.6 mm × 5 µm NST) as a stationary phase and a mobile phase consisting of acetonitrile and a sodium acetate buffer 0.7 mM, pH = 5.7 (65:25 (*v/v*)). The method underwent validation following verification parameters set by the Food and Drug Administration (FDA) [51], with a detection threshold of 2.5 µg/mL of plasma. In addition to this, the approach demonstrated precision and accuracy, covering a linear range of 5–1000 µg/mL of plasma.

The curves were generated by plotting the plasma concentration against time, specifically utilizing the area under the curve from time zero to infinity (AUC^0–∞^). The estimation of these curves was performed using the trapezoid approach [52]. Pharmacokinetic parameters were determined by analyzing plasma concentrations. To assess the disparities between the control + CEF and LPS + CEF groups, various measures were employed to determine bioavailability (AUC), distribution (Vd), and elimination (t_1/2_ and clearance).

The PKSolver add-in in Microsoft Excel^®^ was used to perform pharmacokinetic analysis [53].

### 4.7. Statistical Analysis

In the statistical analysis of pharmacokinetics, where data were given as a median, tests were conducted using statistical tests performed with Statistica 7.0 (StatSoft Power Solutions, Inc., Hamburg, Germany); the significance threshold for all statistical tests was set at 5%. The statistical analysis employed in this study utilized the two-tailed Mann–Whitney test for unpaired data. The same statistical test was used to quantify nitrite levels across the different study groups.

The VEGF quantification and respiratory mechanics statistical analysis was conducted using Prism (version 7.0; San Diego, CA, USA). The results were presented as the mean value and the standard error (mean ± S.E.). The statistical analysis involved using two-tailed Mann–Whitney tests for unpaired data or two-way analysis of variance (two-way ANOVA), followed by a Tukey’s test for conducting multiple comparisons. A significance threshold of 5% or less (*p* ≤ 0.05) and a confidence interval of 95% were utilized for all tests.

## Figures and Tables

**Figure 1 antibiotics-13-00083-f001:**
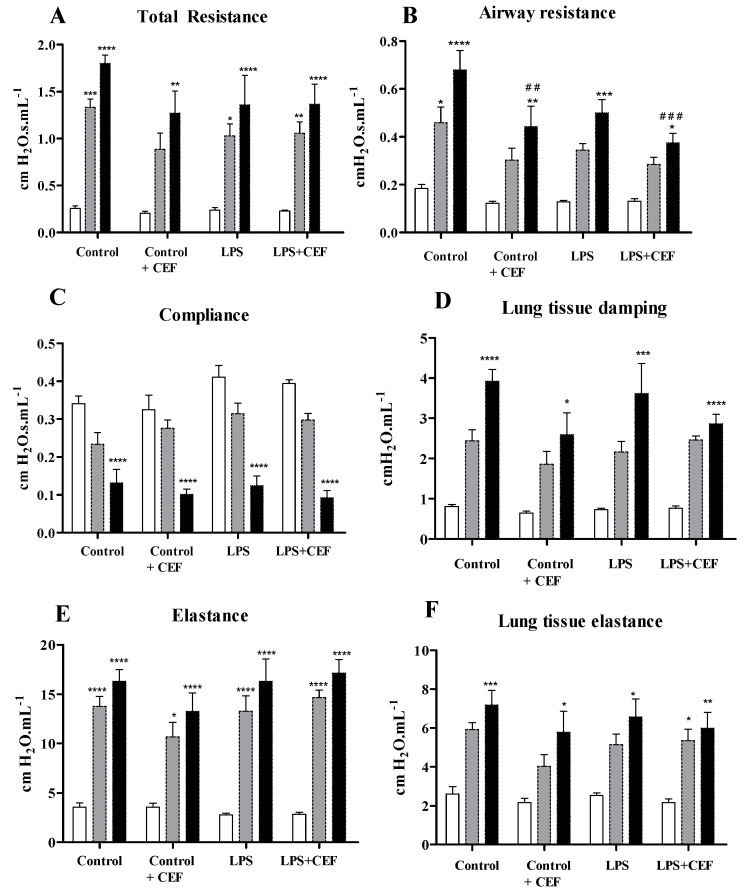
Respiratory mechanic values are expressed as mean ± SEM. In blank saline administration, in grey MCh50 and black for MCh100. Total resistance (**A**), Airway resistance (**B**), Compliance (**C**); Lung tissue damping (**D**), Elastance (**E**); and Lung tissue elastance (**F**), in rats of the control group, control + ceftriaxone (control + CEF) group, LPS group and LPS + CEF group (n = 5−8). * for saline vs. MCh50 or MCh100 within the same group (* *p* < 0.05; ** *p* < 0.01; *** *p* < 0.001; **** *p* < 0.0001). # for comparison of MCh100 of control group vs. MCh100 of different groups (## *p* < 0.01; ### *p* < 0.001).

**Figure 2 antibiotics-13-00083-f002:**
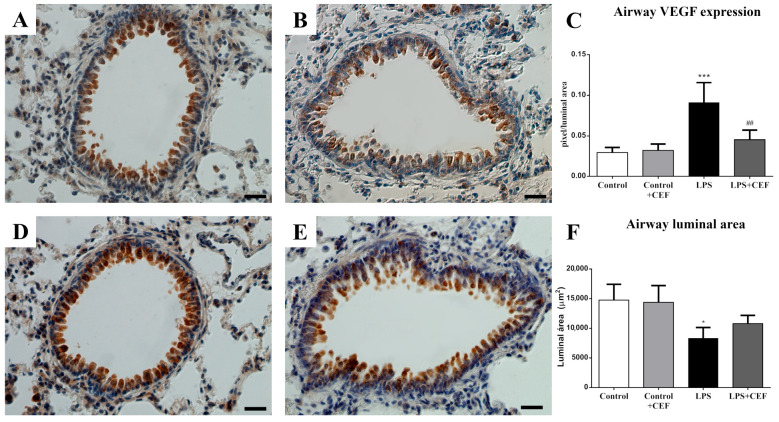
Morphological analyses of airway VEGF expression in rats of control group (**A**), control + ceftriaxone (control + CEF) group (**B**), LPS group (**D**), and LPS + CEF group (**E**) after immunohistochemistry for VEGF/DAB-peroxidase/Harris Haematoxylin, Scale = 25 μm. Morphometric analyses of airway VEGF expression (**C**) and luminal area (**F**) of all experimental groups expressed as mean ± SEM (n = 5). * for control vs. different groups (* *p* < 0.05; *** *p* < 0.001. # for comparison of LPS group vs. LPS + CEF group (## *p* < 0.01).

**Figure 3 antibiotics-13-00083-f003:**
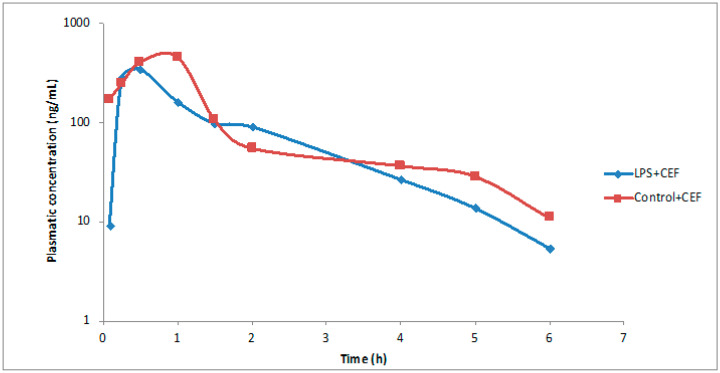
Plasma concentration curves versus Time of ceftriaxone in the control + CEF and LPS + CEF groups (represented by the average of data obtained).

**Table 1 antibiotics-13-00083-t001:** Bronchoalveolar lavage: inflammatory parameters. Values expressed as mean ±SD 24 h after endotoxin administration and 1 h after treatment with ceftriaxone.

	WBC (Cells·10^5^/mL)	Macrophage (%)	Lymphocyte (%)	Neutrophil (%)	Foamy Macrophages (%)	NOx (µM)
Control	456.56 ± 9.70	48.38 ± 2.48	35.34 ± 1.88	0.44 ± 0.12	2.86 ± 0.56	2.31 ± 0.26
Control + CEF	842.81 ± 19.75 **	43.84 ± 3.50 **	52.83 ± 3.60 **	0.38 ± 0.07	2.03 ± 0.18 *	3.59 ± 0.17 *
LPS	1135.00 ± 15.56 ***	61.52 ± 1.77 *	44.46 ± 3.06 *	0.88 ± 0.33 *	3.86 ± 0.54 ^#^	15.65 ± 0.61 *^###^
LPS + CEF	1194.68 ± 7.0 *	53.96 ± 2.37 *	43.89 ± 2.58 **	2.25 ± 0.36 **^##^	1.87 ± 0.28 ^&^	6.82 ± 0.50 *^##&&^

* The values are mean ± SEM; control group, control + ceftriaxone (control + CEF) group, LPS group, and LPS + CEF group (n = 8). * For comparison with control group (* *p* < 0.05; ** *p* < 0.01; *** *p* < 0.001). ^#^ For comparison with control + CEF group (^#^ *p* < 0.05; ^##^ *p* < 0.01; ^###^ *p* < 0.001). ^&^ for comparison with the LPS group (^&^ *p* < 0.05; ^&&^ *p* < 0.001). WBC = white cells.

**Table 2 antibiotics-13-00083-t002:** Estimated pharmacokinetic parameters after IP administration of ceftriaxone (dose = 100 mg/kg) in control and sepsis-induced rats by LPS (n = 8 for each profile). Data expressed as median, CI = 95%.

Parameter	Control + CEF	LPS + CEF	*p* Value
AUC^0–∞^ (h × µg/mL)	672.47 *; 634.31–710.63	514.01; 485.43–542.60	0.0002
t_½_ (h)	2.29 *; 0.67–3.91	1.13; 1.10–1.16	0.0140
Cl_T_/f (mL/h/kg)	149.31 *; 140.81–157.81	195.30; 184.49–206.11	0.0002
Tmax (h)	0.5332 *; 0.51–0.55	0.3600; 0.34–0.37	0.0002
Cmax (µg/mL)	357.50 *; 330.98–384.02	300.13; 284.30–315.95	0.0002

* Mann–Whitney test, (*p* < 0.05). AUC^0–∞^—Area under the plasma concentration curve versus time from zero to infinity; Tmax: time to reach maximum plasma concentration; ClT/f: total apparent clearance; Cmax—plasma concentration maximum; t_1/2_—half-life.

## Data Availability

The data presented in this study are available on request from the corresponding author.

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
