# Peer review of "Endotoxin-Induced Sepsis on Ceftriaxone-Treated Rats’ Ventilatory Mechanics and Pharmacokinetics"

_antibiotics, 2024, doi:10.3390/antibiotics13010083_

Round 1

Reviewer 1 Report

Comments and Suggestions for Authors

Study by Juliana Savioli Simões et al “Endotoxin-induced sepsis on ceftriaxone-treated rats' ventilatory mechanics and pharmacokinetics”.

 The authors provided evidence for ventilatory mechanics and pharmacokinetics on Endotoxin-induced sepsis on ceftriaxone-treated rats. The authors used male Wistar rats with control and sepsis induced by endotoxin LPS. Control received only intraperitoneal saline or saline + ceftriaxone (CEF) and treated groups ceftriaxone (100mg/Kg) IP. Authors then evaluated respiratory mechanics, nitrite, and VEGF quantification for cell evaluation. For pharmacokinetic evaluation, two groups received ceftriaxone. Respiratory mechanics shows a decrease in total airway resistance, dissipation of viscous energy, and elastance of lung tissues in all sepsis induced groups compared to control. Vascular endothelial growth factor VEGF and NOx values were higher in sepsis animals compared to control, and ceftriaxone was able to reduce both parameters. The pharmacokinetic parameters for ceftriaxone, such as bioavailability, absorption, and terminal half-life, were smaller in the sepsis-induced group, compared to than in the control group.

In conclusion authors found that ceftriaxone showed a reduction of resistance in the airways. In addition, CEF lowers nitrite levels in the lungs and acts on their adverse effects, reflecting pharmacological therapy of the disease.

 In the introduction section authors explained the process of sepsis and progressive deterioration of acute lung injury (ALI) and ARDS as a significant clinical problem causing severe edema, hypoxemia, and increased lung density. Ceftriaxone with a potent antimicrobial activity against the common Gram-negative and Gram-positive bacteria. The stability and relatively long elimination half-life of ceftriaxone (approx-7h) make this cephalosporin a good choice for the treatment of patients with a number of several infections. However, authors mentioned several factors, with the selection of antimicrobial therapy and the correct dose being key determinants. Ceftriaxone, a third-generation cephalosporin, has been approved by the World Health Organization (WHO) as the primary antibiotic of choice for the management in managing patients admitted to ICU.

 Infections are the most common etiology of ARDS and cephalosporin was used to improve the disease outcome. Authors hypothesized that ARDS could cause physiological changes that can alter treatment efficacy of treatment. Therefore, in this study, authors integrated an investigation of the pharmacokinetics of ceftriaxone and its impact on pulmonary function, aiming to enhance comprehension of the treatment effectiveness in ARDS.

 The authors studied the CEF consequences on the lungs after one dose administration and one hour of acute activity close to the maximum plasma Concentration (Cmax) of CEF in rats. Authors found that even in just one hour, the CEF improves the pulmonary functions shown by Raw. Authors found an increase in macrophages in the LPS group, CEF group has lower macrophage levels than the LPS group, e lymphocyte values increase in levels in the LPS groups, decrease in values of foam cells in CEF group compared to the other groups. Neutrophil levels were higher in the groups treated with endotoxin. The levels of the inflammatory marker NOx were significantly higher in all groups. Nitrite levels in the LPS+CEF group were significantly lower than in the LPS group. The expression of VEGF is higher in animals in the LPS group compared to the other groups. The LPS group exhibited a decrease in the airway luminal area as compared to the Control group which was subsequently repaired by ceftriaxone treatment. In the pharmacokinetic parameters the LPS+CEF group exhibited a statistically significant reduction in the area under the curve parameter compared to the treated animals (Control+CEF).

 Sepsis patients may have capillary permeability and endothelial lesions in veins and microcirculations due to VEGF production. Prolonged mechanical respiration increases oxygen levels and reduces VEGF in glandular cells and alveolar epithelial tissue, decreasing pathology-related VEGF production. Authors found that CEF decreased VEGF in the LPS+CEF and control groups.

Finally, authors found that animals developed sepsis after 24 hours. Ceftriaxone reduced resistance, improving airflow, reduce lung VEGF expression and nitrite levels, reduces infection, and suggests pharmacological action of CEF. Authors indicated that the indicators improved despite decreased CEF bioavailability for septic animals.

 The study seems fine, methods are written in detail, data are convincing, article language is adequate throughout the manuscript. There are a few corrections needed throughout the manuscripts which are easy to correct such as:

Line – 15 – LPS full form missing.

Line 16 – ‘received’ after group.

Line 18 – VEGF full form missing.

 Study by Juliana Savioli Simões et al “Endotoxin-induced sepsis on ceftriaxone-treated rats' ventilatory mechanics and pharmacokinetics”. The study has clinical significance in treating sepsis with CEP and its ventilatory mechanics and pharmacokinetics.

Comments on the Quality of English Language

The study seems fine, methods are written in detail, data are convincing, article language is adequate throughout the manuscript. There are a few corrections needed throughout the manuscripts which are easy to correct.

Line – 15 – LPS full form missing.

Line 16 – ‘received’ after group.

Line 18 – VEGF full form missing.

Author Response

Dear Reviewer

We are grateful for the opportunity to improve our work. 

The paper was revised according to the comments of all reviewers, and the answers are below.

Reviewer: 1

Line 15 – LPS full form missing.

We added the full form

Line 16 – ‘received’ after group.

The word was added

Line 18 – VEGF full form missing.

We added the full form

Reviewer 2 Report

Comments and Suggestions for Authors

CRITIQUE

Endotoxemia, induced by intraperitoneal injection of bacterial lipopolysaccharide, induces a potentially lethal auto-inflammatory response characterized by high blood levels of TNF, IL-1, IL-6, MIF, HMGB1, and other pro-inflammatory cytokines in concert with organ dysfunction.  For example, it is well-established from numerous animal studies and clinical observations that NO production through iNOS is involved in the lung injury due to endotoxemia. Endotoxemia in rats by intravenous administration of LPS causes systemic hypotension, endothelial damage and acute lung injury accompanied by increased expression of mRNA for iNOS, TNF-α and IL-1β [J Biomed Sci. 1999;6:28–35].

Furthermore, lung vascular leakage and inflammatory lung injury in response to pro-inflammatory cytokine-pulmonary endothelial cell interactions induced by LPS is widely accepted [World J Crit Care Med. 2012;1(2):50–60]. Also, VEGF was originally identified as a vascular permeability factor and is strongly implicated in the pathogenesis of acute lung injury/ARDS [Respiration. 2014;87(4):329-42].

More relevant than VEGF, HMGB1 and histones are abundantly studied as important mediators of systemic inflammation, complement and coagulation activation, endothelial injury and organ dysfunction that characterize various critical illnesses, such as sepsis [Front Immunol. 2021;11:601815; Front Immunol. 2021;12:650184].

The authors hypothesize that endotoxin-induced ARDS in rats is associated with an underlying pathophysiology that may modify the pharmacokinetics and hence the efficacy of ceftriaxone, a third-generation cephalosporin recommended for primary treatment of Gram-negative or Gram-positive sepsis in humans.  The authors further examine whether ceftriaxone affects pulmonary inflammation and function, separate, evidently, from LPS-mediated effects on airways and pulmonary circulation functions.

The authors show that LPS significantly affected pharmacokinetics of ceftriaxone, for example, reducing both the time required to reach the maximum plasma concentration (Tmax) and the maximum plasma concentration (Cmax)(page 8, lines 220-221).  The authors demonstrate that LPS increases VEGF expression in airways, which was significantly reduced by CEF.  The combined effects of these results are not altogether clear.

Enthusiasm for the manuscript is further diminished by marginal relevance to human disease.  Notably, humans are as much as 100,000-fold more sensitive to LPS than rats, and, arguably, the possibility must be acknowledged that results of experiments in rodents using LPS or other stimulatory microbial molecules might not be applicable to humans. [J Infect Dis. 2010;201(2):175–177.]. Moreover, bolus intravenous injection of endotoxin does not mimic Gram-negative bacterial disease, which is characterized in part by intermittent and low-grade release of endotoxin.  Intraperitoneal injection of LPS, as used in the authors’ model may be, however, a closer approximation to human endotoxemia from human Gram-negative sepsis. Cecal ligation and puncture is perhaps an even more relevant model.

SPECIFIC COMMENTS

Page 2, lines 49-53: Combine the information in this paragraph with the paragraph further down (page 2, lines 66-74).

Page 4, Figure 1: Include PaO2/FiO2 for various groups.

Page 8: The LPS-induced lung injury can be modulated by iNOS-nonspecific and iNOS-specific inhibitors such as N-monomethyl-L-arginine, L-NAME, aminoguanine and dexamethasone.  Consider confirming specificity of this LPS-induced lung injury model.

Page 8, line 218: The statement, “Increased clearance due to infection...” is incorrect.  Experimental animals in this study were not infected.

Page 9, line 224: The statement, “LPS-induced sepsis in rats...” is misleading.  It would be more accurate to indicate that the effects of LPS on pulmonary function after CEF administration were analyzed.

Page 11, line 324: Repeats line 317

Page 11, lines 318 -319: Please clarify why two different doses of LPS (1 mg/kg and 100 mg/kg) were used.

Page 11, line 347:  Animals were ventilated with 0.6 mL/100 g (1.5 mL +/- 0.3 mL) at 110 breaths/min (MV = 165 mL/min).  At this rate, (and presumed I:E = 1:1) ventilated animals had approximately 0.45 sec in exhalation. What assurances (what measure) can the authors provide that there was no air trapping at these vent settings.

Page 11, line 347: What was the FiO2?  Please include with other vent settings

Comments on the Quality of English Language

Completely acceptable.  Only minor corrections need.

Author Response

We are grateful for the opportunity to improve our work. 

The paper was revised according to the comments of all reviewers, and the answers for reviewers are below.

Reviewer: 2

General comments

Endotoxemia, induced by intraperitoneal injection of bacterial lipopolysaccharide, induces a potentially lethal auto-inflammatory response characterized by high blood levels of TNF, IL-1, IL-6, MIF, HMGB1, and other pro-inflammatory cytokines in concert with organ dysfunction.  For example, it is well-established from numerous animal studies and clinical observations that NO production through iNOS is involved in lung injury due to endotoxemia. Endotoxemia in rats by intravenous administration of LPS causes systemic hypotension, endothelial damage and acute lung injury accompanied by increased expression of mRNA for iNOS, TNF-α and IL-1β [J Biomed Sci. 1999;6:28–35].

Furthermore, lung vascular leakage and inflammatory lung injury in response to pro-inflammatory cytokine-pulmonary endothelial cell interactions induced by LPS is widely accepted [World J Crit Care Med. 2012;1(2):50–60]. Also, VEGF was originally identified as a vascular permeability factor and is strongly implicated in the pathogenesis of acute lung injury/ARDS [Respiration. 2014;87(4):329-42].

More relevant than VEGF, HMGB1 and histones are abundantly studied as important mediators of systemic inflammation, complement and coagulation activation, endothelial injury and organ dysfunction that characterize various critical illnesses, such as sepsis [Front Immunol. 2021;11:601815; Front Immunol. 2021;12:650184].

The authors hypothesize that endotoxin-induced ARDS in rats is associated with an underlying pathophysiology that may modify the pharmacokinetics and hence the efficacy of ceftriaxone, a third-generation cephalosporin recommended for primary treatment of Gram-negative or Gram-positive sepsis in humans.  The authors further examine whether ceftriaxone affects pulmonary inflammation and function, separate, evidently, from LPS-mediated effects on airways and pulmonary circulation functions.

The authors show that LPS significantly affected pharmacokinetics of ceftriaxone, for example, reducing both the time required to reach the maximum plasma concentration (Tmax) and the maximum plasma concentration (Cmax)(page 8, lines 220-221).  The authors demonstrate that LPS increases VEGF expression in airways, which was significantly reduced by CEF.  The combined effects of these results are not altogether clear.

Enthusiasm for the manuscript is further diminished by marginal relevance to human disease.  Notably, humans are as much as 100,000-fold more sensitive to LPS than rats, and, arguably, the possibility must be acknowledged that results of experiments in rodents using LPS or other stimulatory microbial molecules might not be applicable to humans. [J Infect Dis. 2010;201(2):175–177.]. Moreover, bolus intravenous injection of endotoxin does not mimic

Response to General Comments

We are grateful for the opportunity to improve our article. The articles/ideas pointed out by the reviewer in the five opening paragraphs. However, our study is concerned with LPS-induced ARDS, and the dates are obtained in BALF. In this case, we try to focus on the lung after LPS + CEF administrations.

Based on reviewer comments, we decided to include in the discussion section an additional text about LPS and VEGF relations. We included: “Current studies established a clear relation between LPS administration and VEGF expression in the lung (Zhang et al. 2020; Tsikis et al., 2022) in the long term. These authors indicate the VEGF impairs could be a possible target to damage lungs induced by LPS instillation in mice. Our study shows an increase in VEGF expression in the airway, however, we also demonstrated a reduced luminal area at a short time.”

- Zhang Z, Lu DS, Zhang DQ, Wang X, Ming Y, Wu ZY. Targeted Antagonism of Vascular Endothelial Growth Factor Reduces Mortality of Mice with Acute Respiratory Distress Syndrome. Curr Med Sci. 2020 Aug;40(4):671-676. doi: 10.1007/s11596-020-2236-7.

- Tsikis ST, Fligor SC, Hirsch TI, Pan A, Yu LJ, Kishikawa H, Joiner MM, Mitchell PD, Puder M. Lipopolysaccharide-induced murine lung injury results in long-term pulmonary changes and downregulation of angiogenic pathways. Sci Rep. 2022 Jun 17;12(1):10245. doi: 10.1038/s41598-022-14618-8. 

Also, about the LPS model for inducing lung injury, the reviewer is correct. However, it was the choice model to develop our protocol. We carried out using intraperitoneal lipopolysaccharide (LPS) administration to induce lung injury based on many studies in literature, as we see in the listed references in the present work. We believe that this procedure is the most commonly utilized murine model of ALI and shares similarities in pathophysiology to human ARDS as mentioned by Matute-Bello et al., 2008.

  • Matute-Bello, G., Frevert, C. W. & Martin, T. R. Animal models of acute lung injury. Am. J. Physiol. Lung Cell Mol. Physiol. 295, L379-399. https://doi.org/10.1152/ajplung.00010.2008 (2008).

SPECIFIC COMMENTS

Page 2, lines 49-53: Combine the information in this paragraph with the paragraph further down (page 2, lines 66-74).

Response: The information was combined and both combined paragraphs start in line 61

Page 4, Figure 1: Include PaO2/FiO2 for various groups.

Response: Unfortunately, our protocol was organized to prioritize lung functions. The flexiVent equipment was used to assess lung function and FIO2 was not provided by the equipment. The flexiVent is widely considered the gold standard for in vivo measurements of respiratory mechanics in small animals. It goes beyond the traditional mechanics of resistance and compliance of lung ventilation and captures crucial details about the mechanical properties of the conducting airways, terminal airways, and parenchyma.

No arterial cannulation was performed to obtain arterial gases.

Page 8: The LPS-induced lung injury can be modulated by iNOS-nonspecific and iNOS-specific inhibitors such as N-monomethyl-L-arginine, L-NAME, aminoguanine and dexamethasone.  Consider confirming specificity of this LPS-induced lung injury model.

Response: In the present study we decided to measure the nitrite/nitrate concentration (NOx) and not measure the NO concentration in the BALF after LPS administration. To clarify the explanation, we included the following text: “LPS is a major component of the outer membrane of gram-negative bacteria and is known to be a key pathogenic stimulator for multiple organ dysfunction. In the case of sepsis, circulating LPS as a pathogen-associated molecular pattern (PAMP) stimulates the innate immune system and mediates local or systemic inflammation.”

This way, we believed the LPS-induced ARDS and NOx in the BALF are increased, suggesting that was triggered by the inducible NO synthase (iNOS), as mentioned by Su et al. 2012

(Su CF, Kao SJ, Chen HI. Acute respiratory distress syndrome and lung injury: Pathogenetic mechanism and therapeutic implication. World J Crit Care Med. 2012 Apr 4;1(2):50-60. doi: 10.5492/wjccm.v1.i2.50. PMID: 24701402).

However, isoforms modulated in the lung were not evaluated in our study.

Page 8, line 218: The statement, “Increased clearance due to infection...” is incorrect.  Experimental animals in this study were not infected.

Response: The sentence was altered to: “Increased clearance in LPS+CEF group…and is actually in line 217.

Page 9, line 224: The statement, “LPS-induced sepsis in rats...” is misleading.  It would be more accurate to indicate that the effects of LPS on pulmonary function after CEF administration were analyzed.

Response: The sentence was modified starting at line 223.

Page 11, line 324: Repeats line 317

Response: The second sentence was removed

Page 11, lines 318 -319: Please clarify why two different doses of LPS (1 mg/kg and 100 mg/kg) were used.

Response: The original text correctly described the doses. They were: LPS 1mg/kg and CEF 100mg/kg. Then, the first dose is for LPS, the second is for CEF.

Page 11, line 347:  Animals were ventilated with 0.6 mL/100 g (1.5 mL +/- 0.3 mL) at 110 breaths/min (MV = 165 mL/min).  At this rate, (and presumed I:E = 1:1) ventilated animals had approximately 0.45 sec in exhalation. What assurances (what measure) can the authors provide that there was no air trapping at these vent settings.

Response: The reviewer can be right, but in concordance with our expertise, the ventilation parameters setting was carried out using the flexiVent (too small animals) 6mL/kg to avoid lung damage during ventilation measurements (to detail see ref. 25).

Page 11, line 347: What was the FiO2?  Please include with other vent settings

Response: Unfortunately, our protocol was organized to prioritize lung functions. The flexiVent equipment was used to assess lung function and FIO2 was not provided.

Best regards

Vanessa Bergamin Boralli

Round 2

Reviewer 2 Report

Comments and Suggestions for Authors

The authors have satisfactorily clarified issues identified, and have completely addressed questions raised in the first review of their manuscript.  Thank you.